# Batched First-Order Methods for Parallel LP Solving in MIP

Nicolas Blin [1]   Stefano Gualandi [2]   Christopher Maes [1]   Andrea Lodi [3]   Bartolomeo Stellato [4]

## Abstract

We present a batched first-order method for solving multiple linear programs in parallel on GPUs. Our approach extends the primal-dual hybrid gradient algorithm to efficiently solve batches of related linear programming problems that arise in mixed-integer programming techniques such as strong branching and bound tightening. By leveraging matrix-matrix operations instead of repeated matrix-vector operations, we obtain significant computational advantages on GPU architectures. We demonstrate the effectiveness of our approach on various case studies and identify the problem sizes where first-order methods outperform traditional simplex-based solvers, depending on the computational environment. This is a significant step toward integer programming algorithms that tightly exploit GPU capabilities. We argue that some specific operations should be allocated to GPUs and performed in full, instead of relying on lightweight heuristic approaches on CPUs.

## 1. Introduction

First-order methods for solving linear programming (LP) problems have recently gained traction for the (approximate) solution of very large instances. This is because first-order methods are a natural choice for parallelization, which is crucial for exploiting GPUs.

There has been significant progress in solving LPs with first-order methods, starting from the first GPU implementations of operator-splitting solvers based on the alternating direction method of multipliers (ADMM), such as the SCS solver (O'Donoghue et al., 2016; O'Donoghue, 2021), the OSQP solver (Stellato et al., 2020; Schubiger et al., 2020), and, more recently, the primal-dual hybrid gradient (PDHG) algorithm (Chambolle & Pock, 2011) implementation in the PDLP solver (Applegate et al., 2021; 2023; Lu & Yang, 2025). So far, the target of this research area has been solving extremely large LPs, and there is some skepticism that this technology can be applied in mixed-integer programming (MIP) (Rothberg, 2024). Indeed, interior point algorithms are also effective and parallelizable, so the LP relaxation of any MIP is reasonably tractable, and the simplex method seems unbeatable in its ability to warm-start after small modifications such as a variable bound change or a round of cuts.

GPU use in MIP has also been limited by the technological burden of moving large amounts of data from CPUs (where the computation has traditionally been run) to GPUs and vice versa. This difficulty has been observed, for example, in the context of using machine learning (ML) models to augment branch-and-bound methods (Scavuzzo et al., 2024). Significant success has been obtained (see, *e.g.*, (Bonami et al., 2022; Berthold et al., 2025)), but the CPU/GPU interaction has slowed the adoption of ML models using neural network (NN) representations, especially when those models need to be run at each node of the branch-and-bound tree, for example for variable selection in branching (Gasse et al., 2019; Gupta et al., 2020).

In other words, effective GPU exploitation in MIP, beyond improvements in GPU memory, may require rethinking the branch-and-bound method itself: potentially abandoning some of its foundational components (the simplex method) and removing its computational work limits.

Along this path, this paper shows that one GPU benefit for MIP is the ability to execute, in full and on GPUs, chunks of the computation that traditionally are performed by clever heuristics that approximate the work and limit the computational load. Two such chunks are strong branching and optimization-based bound tightening (OBBT), which involve solving batches of LPs that differ only in their variable bounds. State-of-the-art implementations of batched LP solutions arise in differentiable optimization, where problems are structured as neural network layers (Agrawal et al., 2019; Lu et al., 2024; Besançon et al., 2024) and primarily use basic vectorization operations (*e.g.*, vmap). These

[1]NVIDIA [2]Department of Mathematics, University of Pavia, Italy [3]Jacobs Technion-Cornell Institute, Cornell Tech, New York, USA [4]Department of Operations Research and Financial Engineering, Princeton University, USA. Correspondence to: Bartolomeo Stellato <bstellato@princeton.edu>.

*Proceedings of the 43rd International Conference on Machine Learning*, Seoul, South Korea. PMLR 306, 2026. Copyright 2026 by the author(s).

methods do not fully exploit the parallelism of GPU-based matrix operations. They also require data duplication across batch instances, since many solvers lack full thread safety for parallel instance solving.

In this paper, we propose a fully parallelizable first-order method directly designed to solve batches of LPs all at once on GPUs, overcoming limitations of existing batch approaches. In the case of strong branching, this allows a precise initialization of pseudocosts (the fine-grained information MIP solvers use for guiding branching) in only a few rounds of memory communication, gaining quality without paying a high price for CPU/GPU interaction.

The remainder of the paper is organized as follows. In Section 2, we review the relevant literature concerning first-order methods for LPs and their GPU implementations. In Section 3, we present the way we formulate an LP so as to be amenable to the first-order algorithm described in Section 4. In Section 5, we discuss our contribution in solving batches of LPs by first-order methods. In Section 6, we present our computational experiments with our implementation of BATCHLP to perform strong branching and OBBT on standard benchmarks from the literature. Finally, in Section 7, we draw some conclusions.

**Conflict of Interest Disclosure.** Nicolas Blin and Christopher Maes are employed by NVIDIA, which leads the development of cuOpt. The cuOpt dual simplex is used as the CPU baseline throughout this paper, and BATCHLP is integrated into the cuOpt branch-and-bound framework in Section 6.

## 2. Literature review

The first effective first-order method for LP is the PDLP algorithm (Applegate et al., 2021; 2023). PDLP applies the primal-dual hybrid gradient (PDHG) method, see (Chambolle & Pock, 2011), and uses restarting and averaging to accelerate it. Several other enhancements were included in PDLP such as adaptive step-size (see also (Chambolle et al., 2024)) and problem scaling.

Building on PDLP, the Halpern Peaceman-Rachford (HPR) method takes a weighted average between the current PDHG iterate and an initial point (Lu & Yang, 2024). The initial point is updated when certain restart conditions are met. This leads to an algorithm called restarted Halpern PDHG (rHPDHG). A similar restarted Halpern algorithm is developed in (Chen et al., 2026) with a relaxation step that results in a longer step size. Inspired by (Chen et al., 2026), an extension is considered in (Lu & Yang, 2024), called reflected restarted Halpern PDHG ($r^2$HPDHG), where the Halpern iteration is performed on the reflection of the PDHG operator instead of the operator itself. Finally, the extension

of rHPDHG to conic LP is presented in (Xiong & Freund, 2024).

A geometric interpretation of PDLP is presented in (Liu & Lu, 2026) and used to design a new crossover algorithm that recovers a vertex solution for an LP. Infeasibility detection (*i.e.*, recognizing infeasible subproblems to avoid useless iterations) plays a key role in these algorithms, and is extensively explored in (Applegate et al., 2024; Banjac et al., 2019).

The idea of extending a first-order method to solve batches of LPs on the GPU for strong branching was first introduced in (Nair et al., 2020). The goal and methodology of (Nair et al., 2020) are fundamentally different from our proposal: there, the GPU implementation of strong branching is used to collect data for training an ML model for the branching task, *i.e.*, to approximate strong branching. Instead, in this paper, we propose to use GPUs to do strong branching, not to approximate it, for enough iterations to initialize pseudo-costs. The way GPUs are used for strong branching is also very different. In (Nair et al., 2020), the batches of LPs are solved by a modification of ADMM, and in a setting where the MIP algorithm branches on all variables rather than only on fractional ones. Finally, their speedup assessment against traditional CPU implementations ignores that each LP can be solved by starting from the optimal basis of the simplex method, performing only a few pivots to converge. This leads to an overestimation of the speedup; our methodology assesses it carefully in different benchmark settings, see Section 6.

A separate line of work uses GPUs to accelerate heuristic and combinatorial search itself, by parallelizing the *tree traversal* rather than the per-node work. Notable examples include massively parallel $A^*$ search on a GPU (Zhou & Zeng, 2015) and GPU-accelerated pathfinding (Bleiweiss, 2008), which expand many nodes simultaneously to exploit GPU throughput. Our approach is complementary: we parallelize the *LP solution* itself via batched matrix-matrix operations, leaving the tree-search logic unchanged.

## 3. Problem formulation

Consider a linear optimization problem in the primal-dual form

$$
\begin{array}{llll}
\min & c^T x & \max & -\phi_{[\underline{x}, \bar{x}]}(r) - \phi_{[l,u]}(y) \\
\text{s.t.} & l \leq Ax \leq u, & \text{s.t.} & c + A^T y + r = 0, \\
& \underline{x} \leq x \leq \bar{x}, & & y \in B_{[l,u]}, \quad r \in B_{[\underline{x}, \bar{x}]},
\end{array}
\tag{1}
$$

where the decision variables are $x \in \mathbf{R}^n$, and the coefficients of the linear objective function are $c \in \mathbf{R}^n$. The constraints are defined by matrix $A \in \mathbf{R}^{m \times n}$ and vectors $l$, $u$, $\underline{x}$, and $\bar{x}$, with $l_i, \underline{x}_i \in \{-\infty\} \cup \mathbf{R}$ and $u_i, \bar{x}_i \in \mathbf{R} \cup \{+\infty\}$. We represent equality constraints as $l_i = u_i$ and variable

fixings as $\underline{x}_i = \bar{x}_i$. We define the dual variables for the inequality constraints as $y \in \mathbf{R}^m$ and for variable bounds as $r \in \mathbf{R}^n$. The set $B$ represents the barrier cone of a hyperrectangle, *i.e.*, $B_{[a,b]} = \{v \mid v_i \geq 0 \text{ if } a_i = -\infty; \ v_i \leq 0 \text{ if } b_i = \infty; \ v_i \in \mathbf{R} \text{ otherwise}\}$. The function $\phi$ represents the support function of a hyperrectangle, *i.e.*, $\phi_{[a,b]}(v) = \sup_{z \in [a,b]} v^T z = b^T v_+ + a^T v_-$, where $v_+ = \max\{v, 0\}$ and $v_- = \min\{v, 0\}$, with $\max$ and $\min$ applied elementwise.

To derive our algorithm, we reformulate (1) as the following saddle-point problem

$$\max_y \ \min_{\underline{x} \leq x \leq \bar{x}} \ c^T x + y^T A x - \phi_{[l,u]}(y). \quad (2)$$

# 4. Primal-dual hybrid gradient to solve a single instance

We apply the primal-dual hybrid gradient method (PDHG) (Chambolle & Pock, 2011) with reflected Halpern iterations (Halpern, 1967; Lu & Yang, 2024; Lu et al., 2025) to solve problem (1). The iterations in terms of the primal-dual pair $z^k = (x^k, y^k) \in \mathbf{R}^{n+m}$ consist of

$$z^{k+1} = \frac{k+1}{k+2}\left(2T(z^k) - z^k\right) + \frac{1}{k+2}z^0, \quad (3)$$

with the main operator $T$ being defined as

$$T(z^k) = \left\{ (x, y) \ \middle| \ \begin{aligned} &x = \Pi_{[\underline{x},\bar{x}]}\left(x^k - \tau(c + A^T y^k)\right) \\ &y = y^k + \sigma A(2x - x^k) \\ &\quad - \sigma\Pi_{[l,u]}(\sigma^{-1}y^k + A(2x - x^k)) \end{aligned} \right\}.$$

Here, $z^0$ is the initial iterate used in the Halpern iterations as the anchor point. Moreover, $\Pi_{[a,b]}$ is the Euclidean projection onto a hypercube defined as the elementwise operation $\Pi_{[a,b]}(v) = \max\{\min\{v, b\}, a\}$. The primal and dual step-sizes are $\tau$ and $\sigma$, respectively. As commonly done in PDHG (Applegate et al., 2021), we parametrize the step sizes as $\tau = \eta/w$ and $\sigma = \eta w$, where we refer to $\eta$ as the step size and $w$ as the primal weight.

**Restarts.** Similarly to (Lu et al., 2025), we apply an adaptive restart scheme based on the fixed-point residual progress of the non-reflected iterates. More specifically, we define the fixed-point residual metric

$$r(z) = \|T(z) - z\|_M, \text{ with } M = \begin{bmatrix} (w/\eta)I & A^T \\ A & (1/(\eta w))I \end{bmatrix}. \quad (4)$$

To restart, we partition the iterations in two loops, an outer loop indexed by $n$ and an inner loop indexed by $k$, with corresponding iterates $z^{n,k}$. The algorithm repeats iteration (3) until one of the following restart conditions is met:

- sufficient decay: $r(z^{n,k}) \leq \beta_s r(z^{n,0})$,
- necessary decay and no inner progress: $r(z^{n,k}) \leq \beta_n r(z^{n,0})$ and $r(z^{n,k}) > r(z^{n,k-1})$, and
- iteration limit: $k > \beta_a K$, with $K$ being the total number of iterations.

In this case, the anchor point $z^0$ is set to the current iterate.

**Step-size and primal weight.** We adopt the constant step-size from (Lu et al., 2025), where $\eta = 0.998/\|A\|_2$ and the exponential smoothing technique to update the primal weight $w$ (Applegate et al., 2025). Specifically, whenever $d^n = \|x^{n+1,0} - x^{n,0}\|/\|y^{n+1,0} - y^{n,0}\|$ is finite at every outer loop iteration, we update $w^{n+1} = \exp\left(\theta \log(d^n) + (1-\theta)\log(w^n)\right)$.

**Stopping criteria.** This algorithm, including restarts, has been analyzed in (Lu & Yang, 2024) and shown to converge at a linear rate. We terminate the algorithm if the following optimality conditions are satisfied:

$$\begin{aligned} &|c^T x^k + \phi_{[\underline{x},\bar{x}]}(r^k) + \phi_{[l,u]}(y^k)| \\ &\quad \leq \epsilon(1 + |c^T x^k| + |\phi_{[\underline{x},\bar{x}]}(r^k) + \phi_{[l,u]}(y^k)|) \\ &\|Ax^k - \Pi_{[l,u]}(Ax^k)\|_2 \leq \epsilon(1 + \|Ax^k\|_2) \\ &\|c + A^T y^k + r^k\|_2 \leq \epsilon(1 + \|c\|_2). \end{aligned} \quad (5)$$

Since iteration (3) does not track a reduced cost vector $r^k \in \mathbf{R}^n$, we compute it as $r^k = \Pi_{B_{[\underline{x},\bar{x}]}}(-c - A^T y^k)$ where $\Pi_{B_{[\underline{x},\bar{x}]}}$ is the Euclidean projection on the barrier cone of the variable bounds. In the actual implementation, we replace $\Pi_{B_{[\underline{x},\bar{x}]}}$ with an alternative formulation that tends to be more robust to bounds taking large values, see Appendix A of (Applegate et al., 2025).

**Infeasibility detection.** To detect infeasibility, we apply conditions based on the infimal displacement vector of operator splitting algorithms (Pazy, 1971; Banjac et al., 2019; Applegate et al., 2024), specifically applied to Halpern iterations (Park & Ryu, 2023). Given the primal-dual iterates $x^{k-1}, y^{k-1}$ and reduced cost $r^{k-1}$, we apply the operator $T$ to obtain $(\tilde{x}^k, \tilde{y}^k) = T(x^{k-1}, y^{k-1})$ and $\tilde{r}^k = \Pi_{B_{[\underline{x},\bar{x}]}}(-c - A^T \tilde{y}^k)$. Then, we compute the displacement vectors $\delta x^k = \tilde{x}^k - x^{k-1}$, $\delta y^k = \Pi_{B_{[l,u]}}(\tilde{y}^k - y^{k-1})$, and $\delta r^k = \Pi_{B_{[\underline{x},\bar{x}]}}(\tilde{r}^k - r^{k-1})$. We define the $\epsilon$-approximate primal infeasibility conditions as

$$\begin{aligned} &\phi_{[l,u]}(\delta y^k) + \phi_{[\underline{x},\bar{x}]}(\delta r^k) < 0, \\ &\|A^T \delta y^k + \delta r^k\| \leq \epsilon\left(\phi_{[l,u]}(\delta y^k) + \phi_{[\underline{x},\bar{x}]}(\delta r^k)\right). \end{aligned} \quad (6)$$

Similarly, the $\epsilon$-approximate dual infeasibility conditions are

$$\begin{aligned} &c^T \delta x^k < 0, \ \|\delta x^k - \Pi_{R_{[\underline{x},\bar{x}]}}(\delta x^k)\| \leq \epsilon|c^T \delta x^k|, \\ &\|A\delta x^k - \Pi_{R_{[l,u]}}(A\delta x^k)\| \leq \epsilon|c^T \delta x^k|, \end{aligned} \quad (7)$$

where $\Pi_{R_{[a,b]}}$ is the Euclidean projection onto the recession cone of the hyperrectangle $[a, b]$, which is defined as $R_{[a,b]} = \{v \mid v_i \geq 0 \text{ if } b_i = \infty; \ v_i \leq 0 \text{ if } a_i = -\infty; \ v_i = 0 \text{ otherwise}\}$.

## 5. Solving batches of problems

Instead of solving a single instance of problem (1), we consider solving a *batch* of $N$ instances with the same data matrix $A$, and with varying problem vectors. We stack the problem vectors in the objective matrix $C = [\, c^1 \ \cdots \ c^N \,] \in \mathbf{R}^{n \times N}$, the variable bound matrices $\underline{X} = [\, \underline{x}^1 \ \cdots \ \underline{x}^N \,], \overline{X} = [\, \bar{x}^1 \ \cdots \ \bar{x}^N \,] \in \mathbf{R}^{n \times N}$, and the constraint bound matrices $L = [\, l^1 \ \cdots \ l^N \,], U = [\, u^1 \ \cdots \ u^N \,] \in \mathbf{R}^{m \times N}$. By defining the matrices of primal iterates as $X^k = [\, x_1^k \ \cdots \ x_N^k \,] \in \mathbf{R}^{n \times N}$, dual iterates as $Y^k = [\, y_1^k \ \cdots \ y_N^k \,] \in \mathbf{R}^{m \times N}$, and combined iterates as $Z^k = [\, z_1^k \ \cdots \ z_N^k \,] \in \mathbf{R}^{(n+m) \times N}$ we can rewrite iteration (3) as

$$Z^{k+1} \quad = \frac{k+1}{k+2} \left( 2T(Z^k) - Z^k \right) + \frac{1}{k+2} Z^0, \quad (8)$$

with the operator $T(Z^k) = (X, Y)$ defined componentwise as

$$\begin{cases} X = \Pi_{[\underline{X}, \overline{X}]} \left( X^k - \tau \otimes (C + A^T Y^k) \right) \\ Y = Y^k + \sigma \otimes A(2X - X^k) \\ \quad - \sigma \otimes \Pi_{[L,U]}(\sigma^{-1} \otimes Y^k + A(2X - X^k)), \end{cases} \quad (9)$$

and the projection operators representing the elementwise matrix projection. In addition, we have $w, \tau, \sigma \in \mathbf{R}^N$, since we keep a different value for each subproblem. For the primal weight and step sizes, we define the columnwise scaling $\sigma \otimes X := [\sigma_1 x_1, \ldots, \sigma_N x_N]$ and $\tau \otimes Y := [\tau_1 y_1, \ldots, \tau_N y_N]$, that is, every column of $X$ and $Y$ uses a different step size.

**Restarts.** To adapt the restarting logic to take care of batches of LPs, we define the *averaged* fixed-point residual metric as $\tilde{r}(Z^k) = (1/N) \sum_{j=1}^N r(z_j^k)$, that is, the average residual of the columns of matrix $Z^k$, where each residual is computed as in (4), but using the corresponding values of primal weight $w_j$ and step-sizes $\tau_j$ and $\sigma_j$. Whenever the average residual $\tilde{r}(Z^k)$ satisfies one of the three conditions as the ones for the single-instance residual (4), we restart all the batches at the same time, by recomputing the new primal weights $w_j^{n+1}$ by exponential smoothing, and the step-sizes $\tau_j^{n+1}$ and $\sigma_j^{n+1}$ accordingly, for each problem $j = 1, \ldots, N$.

**Convergence and subproblem consistency.** The sequential Halpern PDHG with restarts of (Lu & Yang, 2024) converges linearly to a primal-dual solution. In our batched

version, each column of $Z^k = (X^k, Y^k)$ evolves independently with its own primal weight and step sizes, and the only coupling across subproblems comes through the averaged restart criterion $\tilde{r}(Z^k)$. When subproblems are consistent in the sense that they share the data matrix $A$ and differ only in the cost or in the bounds, as in our target FSB and OBBT applications, the averaging is benign: every subproblem still inherits the linear-rate convergence of the underlying restarted Halpern iteration. For inconsistent batches the algorithm still converges, but the averaged criterion may trigger restarts that are suboptimal for individual columns. Per-column adaptive step sizes (Section 4) and per-column restart strategies are natural extensions in that regime.

**Stopping criteria and infeasibility detection.** The stopping conditions for our batched algorithm are as in (5), but rewritten in matrix notation. The same apply to infeasibility criteria (6) and (7). Whenever we finish solving a problem in the batch, we permute the iterate matrices so that the final columns correspond to the solved problems; equivalently, we keep track of $\pi = (\pi_1, \ldots, \pi_N)$ as a permutation of the indices, with $X^k = [\, x_{\pi_1}^k \ \cdots \ x_{\pi_N}^k \,]$ and $Y^k = [\, y_{\pi_1}^k \ \cdots \ y_{\pi_N}^k \,]$. In this way, we stop updating those variables and reduce the kernel sizes dynamically.

**Matrix-matrix products and optimal batch size.** The major bottleneck in the batched algorithm is the matrix-matrix products $A^T Y$ and $AX$, which can be significantly accelerated on GPUs. Moreover, the cost of computing $AX^k$ and $A^T Y^k$ is weakly dependent on the batch size (the number of columns in $X^k$ and $Y^k$). We varied the batch size and timed 10 matrix-matrix products of $AX^0$ and 10 matrix-matrix products of $A^T Y^0$ using the constraint matrix $A$ from the MIPLIB benchmark problem `csched007` (Yunes, 2009). Figure 1 shows the time that the CUDA `cuSPARSE` library takes to compute these matrix-matrix products on an RTX 4500 ADA GPU. From batch size 32 to 512 the time taken for these products remains constant at approximately $0.03\,\mathrm{ms}$. After a batch size of 512, the time for these products increases with the batch size. We define the optimal batch size as the one that maximizes the number of matrix-matrix multiplies per second. The optimal batch size, influenced by factors such as the constraint matrix $A$, the GPU, and the software configuration, cannot be predetermined. For a given problem and GPU, we estimate it by timing a few matrix-matrix products before solving the MIP.

**Naive batched implementation.** Although the computational speedups achievable by matrix-matrix multiplies over batches seem very promising, a naive batched implementation of our algorithm could be very inefficient due to data movement between CPU and GPU, which causes significant memory overhead. In the next section, we tailor

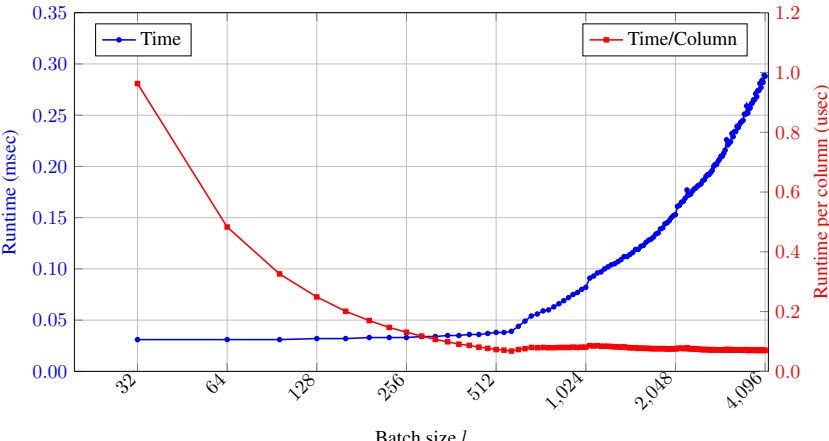

*Figure 1.* Sparse Matrix-Matrix Multiplies vs Batch Size on `csched007`. Total time for 10 $AX$ and 10 $A^T Y$ operations (blue circle markers on the left y-axis) and time/column (red square markers on the right y-axis).

our batched first-order method to strong branching, a core task in mixed-integer programming. In Appendix 5.3, we outline similar customizations to implement optimization-based bound tightening.

### 5.1. Implementation challenges

The transition from sequential to batched PDHG of the main iterations is conceptually intuitive: each sparse matrix-vector product ($Ax$ and $A^T y$) becomes a sparse-dense matrix-matrix product ($AX$ and $A^T Y$), where $X$ and $Y$ are dense. The main iterations store $X$ and $Y$ in row-major layout (and $A$ in CSR format), and the two SpMM operations rely on `cuSPARSE`. The projection operators and other elementwise updates, however, are not covered by `cuBLAS` and require custom CUDA kernels. Several subtle challenges arise; we summarize them next.

**Periodic feasibility and optimality checks.** Every $k$ iterations, we evaluate primal feasibility, dual feasibility, and optimality tolerances independently for each column of $X$ and $Y$. Per-column reductions are most efficient with the dense matrices in column-major layout, so we transpose the row-major iterates to column-major before checking, and back afterwards. The same conversion is used for infeasibility detection.

**Early termination and column removal.** When a subproblem meets its stopping criterion, we extract its primal and dual columns from $X$ and $Y$ and remove them from the active batch. Doing so naively would leave holes in the dense iterates and break the row-major SpMM layout. We instead swap each converged column with a non-converged column from the trailing block, compact the active region, and resume. Performing these swaps one at a time would be prohibitively slow, so we compute the minimum number of

swaps that compact every solved column and execute them in a single batched custom kernel that operates simultaneously on all dense matrices and per-column scalars.

**Batch-level restart strategy.** Restart criteria are more delicate in the batched setting because we want to restart the entire batch simultaneously to preserve the SpMM structure. We use the averaged fixed-point residual $\tilde{r}(Z^k)$ defined above, which couples the restart decision across subproblems. This averaging is benign in our target applications (FSB and OBBT), where the subproblems share $A$ and differ only in cost or bounds, but in general it may restart some subproblems too early and others too late. Per-column restart strategies are a natural extension.

**Warm restart.** When the batch is initialized from a parent LP solution (rather than from scratch), each subproblem inherits a different warm-start point. Handling these heterogeneous initializations is nontrivial because the primal weight $w_j$ and step sizes $\tau_j$, $\sigma_j$ must be set individually for each column while preserving the batched iteration.

**Optimal batch size.** The optimal batch size depends on the instance and on the GPU, and trades off memory usage against the speed of basic operations. The two SpMM operations are expensive, but on larger instances the elementwise updates of $X$ and $Y$ are also significant, so the batch size that maximizes throughput is not always the largest one that fits in memory. As described above, we estimate it at runtime by timing a few SpMM operations before solving the MIP.

### 5.2. Full strong branching

A key step in branch-and-bound algorithms for MIP is choosing which fractional variable to branch on at each node.

Full strong branching (FSB) solves two LPs for each fractional variable: one that rounds up and one that rounds down (Achterberg et al., 2005). The resulting objective values determine the branching score.

Precisely, consider an optimal solution $x^{\text{rel}}$ of the LP relaxation at the current node with $p$ fractional variables. For notational simplicity, assume the first $p$ components of $x^{\text{rel}}$ are fractional. FSB solves $N = 2p$ LPs: for each fractional variable $x_i$ ($i = 1, \ldots, p$), we solve one LP with $x_i \geq \lceil x_i^{\text{rel}} \rceil$ (branch up) and one with $x_i \leq \lfloor x_i^{\text{rel}} \rfloor$ (branch down). All $N$ problems share the same cost vector $c$, constraint bounds $l, u$, and base variable bounds $\underline{x}, \bar{x}$; only one bound component differs per problem.

In matrix notation, the objective matrix is $C = c\mathbf{1}^T \in \mathbf{R}^{n \times N}$ and the constraint bound matrices are $L = l\mathbf{1}^T, U = u\mathbf{1}^T \in \mathbf{R}^{m \times N}$. The variable bound matrices $\underline{X}, \overline{X} \in \mathbf{R}^{n \times N}$ have columns equal to $\underline{x}$ and $\bar{x}$, respectively, except

- Column $i$ (branch up on $x_i$): $(\underline{X})_{ii} = \lceil x_i^{\text{rel}} \rceil$,
- Column $p+i$ (branch down on $x_i$): $(\overline{X})_{i,p+i} = \lfloor x_i^{\text{rel}} \rfloor$.

Since only one bound entry changes per problem, we avoid forming $\underline{X}$ and $\overline{X}$ explicitly; instead, we index into $x^{\text{rel}}$, $\underline{x}$, and $\bar{x}$ as needed. Once memory is allocated for $p$ fractional variables, the same allocation serves smaller values of $p$, which naturally occurs as the tree deepens and more variables become fixed.

### 5.3. Optimization-based bound tightening

Optimization-based bound tightening (OBBT) is a core preprocessing step in MIP solvers, in which we update the bounds of each variable as $\underline{x}_i = \max\{\underline{x}_i, \underline{x}_i^{\text{obbt}}\}$ and $\bar{x}_i = \min\{\bar{x}_i, \bar{x}_i^{\text{obbt}}\}$, where

$$\underline{x}_i^{\text{obbt}} = \begin{array}{ll} \min & x_i \\ \text{s.t.} & l \leq Ax \leq u \\ & \underline{x} \leq x \leq \bar{x} \end{array} \qquad \bar{x}_i^{\text{obbt}} = \begin{array}{ll} \max & x_i \\ \text{s.t.} & l \leq Ax \leq u \\ & \underline{x} \leq x \leq \bar{x}, \end{array}$$

for $i = 1, \ldots, n$. This corresponds to solving $2n = N$ LPs having the same constraint bounds, $l, u$, and variable bounds $\underline{x}, \bar{x}$. The objectives correspond to the maximization and minimization of each component of $x$. In matrix form, this corresponds to constraint bound matrices $L = l\mathbf{1}^T, U = u\mathbf{1}^T \in \mathbf{R}^{m \times N}$, and variable bound matrices $\underline{X} = \underline{x}\mathbf{1}^T, \overline{X} = \bar{x}\mathbf{1}^T \in \mathbf{R}^{n \times N}$. The objective matrix becomes $C = [I \ -I] \in \mathbf{R}^{n \times N}$. As for FSB, we never materialize matrix $C$; instead, we reference its columns directly, as canonical basis vectors with alternating signs.

Before turning to numerical results, we note that none of the MIP solvers, either commercial or academic, uses FSB or performs OBBT extensively on all variables. Work limits keep the computational footprint of these powerful methods under control. For strong branching, one selects a restricted candidate subset of variables and the LPs are solved with a limit on the number of simplex pivots. For OBBT, the LPs are solved heuristically for selected variables, triggered by conditions that estimate the likelihood of strengthening the bound. Section 6 then asks: can one exploit FSB and OBBT at their full strength by moving the associated computation to GPUs?

## 6. Numerical experiments

In this section, we present our computational results for full strong branching and OBBT using benchmark instances from the literature. Specifically, for both strong branching and OBBT, we evaluate the performance of our batched algorithms described in Sections 5.2 and 5.3, respectively, by solving LPs as if we were at the root node of a branch-and-bound tree.

**Implementation details.** We implemented the code in standard C++17 and used the CUDA Toolkit and the Thrust library to develop the main GPU kernels. Our implementation does not perform any dynamic memory allocation at runtime. All necessary data is transferred to the GPU before use according to the techniques described in Section 5.

For the matrix operations, we rely on the `cuSPARSE` and `cuBLAS` libraries. We store in memory both sparse matrices $A$ and $A^T$, and we use `cusparseSpMM` to compute $AX$ and $A^TY$, that is, the product between sparse and dense matrices. We use `cublasDgemv` to compute matrix-vector operations such as $X^Tc$, which yields $c^Tx_i$ for every column $x_i$ of $X$, and `cublasDnrm2` for vector norms. All other functions are executed on the GPU using our own customized CUDA kernels.

**Hyperparameters.** Most hyperparameters of BATCHLP follow the standard defaults of `cuPDLPx` (Lu et al., 2025) and are not tuned per instance. The main hyperparameter that depends on the application is the optimality tolerance: for FSB we use $\epsilon = 10^{-4}$, since branching scores are used only as a heuristic ordering of variables, while for OBBT we use $\epsilon_{\text{dual}} = 10^{-8}$ to guarantee a valid dual bound before tightening any variable bound, as discussed in Section 6 below. BATCHLP remains faster than the multi-threaded dual simplex baseline at both tolerance levels; Table 6 reports a per-instance comparison at $\epsilon = 10^{-4}$ and $\epsilon = 10^{-8}$ on the ORLIB set-covering family.

**Hardware details.** We use two main configurations. The high-end one is a DGX B200 GPU paired with an AMD EPYC 9575F CPU (64 cores) on the `brev` NVIDIA cloud. The lower-end one is a Dell workstation with an RTX 4500 ADA GPU and an Intel Xeon w9-3495X CPU (56 physical cores); this workstation also serves as the multi-threaded

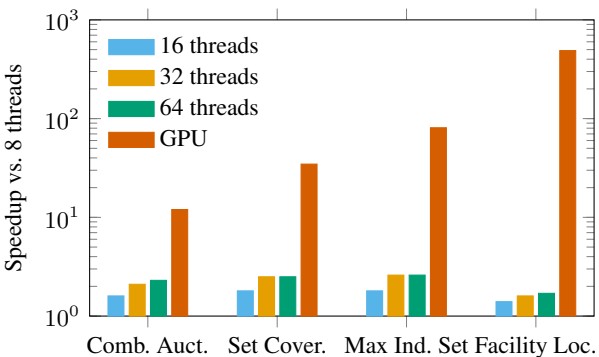

*Figure 2.* Speedup of FSB runtime vs. 8-thread CPU baseline. GPU achieves 12×–488× speedup. Full results in Table 1 (Appendix A).

CPU dual simplex baseline of Table 1 (Appendix A). A cross-hardware comparison on the set-covering family of (Gasse et al., 2019) is reported in Table 7 (Appendix E).

**Comparing CPU and GPU implementations.** Throughout this section, the baseline is the multi-threaded dual simplex of cuOpt 26.02, which runs on *CPU*: despite the name, the dual simplex component is a standard CPU implementation, and we keep it as a baseline because it solves the same preprocessed LPs with the same solver framework as BATCHLP. The comparison is therefore between two LP backends of the same solver: cuOpt CPU dual simplex (multi-threaded) and cuOpt GPU BATCHLP. This reflects the practical deployment scenario in which MIP solvers run on the CPU today, and a GPU is *added* to offload specific subroutines. Our hardware choices represent high-end configurations on both sides, pairing top-tier server CPUs (Intel Xeon, AMD EPYC) with a top-tier GPU (B200). As shown in Table 1 (Appendix A), CPU thread scaling saturates well below ideal: going from 8 to 64 threads yields roughly a 2.5× speedup rather than the ideal 8×, so a wider per-machine GPU advantage is not an artifact of weak CPU baselines.

### 6.1. FSB integration within cuOpt

We integrate BATCHLP into the branch-and-bound method used in NVIDIA cuOpt 26.02 (NVIDIA, 2025). cuOpt performs a round of strong branching after solving the root relaxation to initialize pseudocosts for fractional variables. These pseudocosts are then used to make branching decisions. The default strong-branching implementation in cuOpt splits the subproblems evenly across a set of threads. Each thread uses dual simplex, warm-started from the root's optimal basis, to solve each LP to optimality.

In our main experiments, we use the instances introduced in (Gasse et al., 2019) to perform a computational study on

learning branching strategies that can be competitive with full strong branching, while being computable in a very short time. Here, we take a different perspective: rather than approximating FSB, we aim to execute it by exploiting the GPU, solving each subproblem LP to optimality as fast as possible with BATCHLP.

Figure 2 summarizes the results; full details are reported in Table 1 in Appendix A. Table 1 compares the time required to perform a single round of strong branching using dual simplex on multiple threads (8, 16, 32, and 64 threads on a Dell workstation with 56 physical cores – Intel Xeon CPU) versus BATCHLP. The dual simplex baseline is the multi-threaded implementation of cuOpt 26.02; each thread solves its share of the $|S|$ subproblems sequentially using a dual simplex warm-started from the root LP basis. We use the random generator from (Gasse et al., 2019) to generate instances from four families of combinatorial optimization problems: Combinatorial Auctions, Set Covering, Maximum Independent Set, and Facility Location. Within each family, instances are ordered by increasing size in terms of the number of constraints $m$, variables $n$, and nonzeros $nnz$. The number of subproblems to be solved, denoted by $|S|$, varies across problem classes.

For each family, we report the total speedup relative to the runtime of dual simplex in parallel on 8 threads. The speedup from using more threads is not linear, and the difference between 32 and 64 threads is minimal. In contrast, BATCHLP yields a substantial speedup across all instances (except the smallest one); for the Facility Location MIPs, it achieves a speedup of 488.5×. Complementary results on small and medium-size combinatorial auction and independent set instances from (Gasse et al., 2019), comparing BATCHLP against sequential Gurobi, are reported in Table 5.

### 6.2. OBBT results

In this section, we use BATCHLP independently of cuOpt, customized to perform OBBT by carefully handling the best dual bound found so far on the objective value. We compare its performance to an OBBT implementation based on the commercial solver Gurobi. While in strong branching, the lower numerical accuracy of first-order methods is less critical (branching scores are used only as a heuristic to order variables), in OBBT, we must guarantee the use of a *safe* dual bound before restricting the domain of any variable. For this reason, when running OBBT in BATCHLP, we enforce a higher numerical accuracy when checking dual feasibility ($\epsilon_{\text{dual}} = 10^{-8}$), and we subtract/add

$$\Delta = \epsilon\Big(1 + |c^T x| + \big|\phi_{[\underline{x}, \bar{x}]}(r) + \phi_{[l,u]}(y)\big|\Big),$$

to the lower/upper bounds computed via OBBT. Finally, we update a bound only if the improvement is larger than $10^{-4}$.

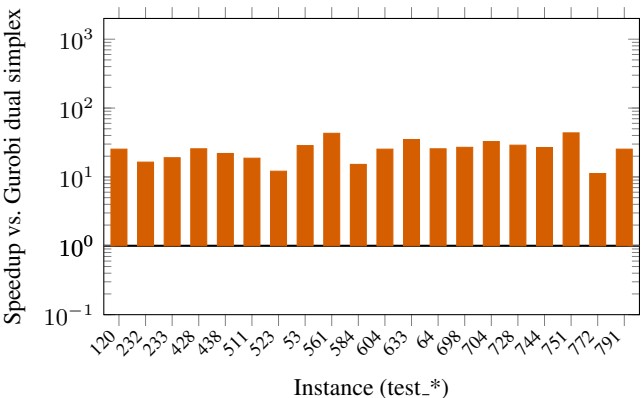

*Figure 3.* OBBT speedup of BATCHLP vs. Gurobi dual simplex on neural network verification instances. Average speedup: 25.7×. Full results in Table 2.

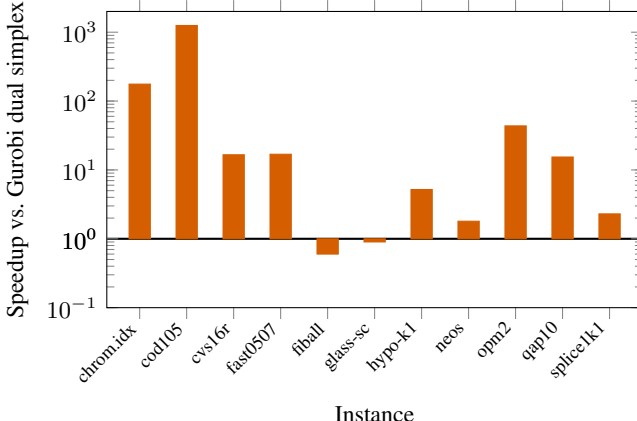

*Figure 4.* MIPLIB 2017 speedup of BATCHLP vs. `cuOpt` dual simplex (28 threads). Values below 1 indicate dual simplex is faster. Full results in Table 3.

In the numerical experiments, we apply OBBT to neural network verification instances from (Nair et al., 2020). Figure 3 summarizes the results, with full details in Table 2. Table 2 reports our results, comparing BATCHLP with a sequential version of Gurobi. As in Table 1, $m$ is the number of constraints and $n$ the number of variables, while $|S| = 2n$ is not explicitly reported. The column `den` is the density of the coefficient matrix $A$ (similar to `nnz` in Table 1). The Gurobi baseline uses Gurobi 13.0 with the dual simplex method (`Method=1`), LP presolve disabled (`Presolve=0`), and a single thread (`Threads=1`); each of the $2n$ OBBT subproblems is solved sequentially with no GPU acceleration. For Gurobi dual simplex, we report the total runtime and the average number of pivots required to solve each subproblem to optimality. For BATCHLP, we report the total runtime, the speedup over the sequential version, and the number of subproblems solved before hitting the iteration limit. The last two columns report the number of variables whose domain changes, and the average domain reduction (as a percentage of the original domain). We remark that, since we perform OBBT on the model obtained after Gurobi MIP presolve, several standard bound-reduction techniques have already been applied to the original model.

**Interpreting the speedup.** The results in Table 2 show a large speedup (25.7 on average), which is significant given the quality of the Gurobi simplex implementation. This average speedup of BATCHLP over sequential dual simplex should be interpreted as follows: under ideal CPU parallelism (which Table 1 shows does not happen), solving the OBBT LPs in parallel with dual simplex using about 26 threads should yield a runtime comparable to that of BATCHLP on a single GPU.

### 6.3. Results on MIPLIB 2017

Finally, we present preliminary results on running BATCHLP on MIPLIB 2017 using the default strong-branching strategy implemented in `cuOpt` 26.02, namely running the dual simplex in parallel with a limit of 200 pivots per subproblem. Figure 4 summarizes the results; full details are reported in Table 3 in the Appendix. Table 3 reports results for a subset of instances on which BATCHLP on an H100 GPU is competitive with the dual simplex running in parallel on 28 threads (i.e., 28 physical cores). The dual simplex baseline is the implementation of `cuOpt` 26.02 running on 28 threads pinned to 28 physical cores, with the default 200-pivot work limit per subproblem. Unsurprisingly, for all other MIPLIB 2017 instances not reported in the table, the dual simplex on 28 threads (with the 200-pivot limit) is faster than BATCHLP.

**When the GPU is slower.** Two MIPLIB instances, `fiball` (0.6×) and `glass-sc` (0.9×), are slower under BATCHLP than under multi-threaded dual simplex. The cause is structural rather than numerical: `fiball` has a wide sparse coefficient matrix ($n/m \approx 13.8$) and only 518 subproblems, so the SpMM operations are memory-bound and the batching benefit is limited; `glass-sc` has a tall dense matrix ($m/n \approx 28.6$) and only 202 subproblems, a regime that favors direct methods. We note that `glass-sc` still achieves a 9.4× speedup against *sequential* Gurobi; the apparent slowdown in Table 3 reflects multi-threaded CPU parallelism closing the gap rather than a regression of BATCHLP. These cases help characterize when GPU offloading is most beneficial: large $|S|$ and moderate-density coefficient matrices. To isolate the contribution of the batching mechanism from the contribution of running PDLP on a GPU, Table 4 in Appendix C compares sequential PDLP with BATCHLP on the same GPU; speedups range from

$1.2\times$ to $160\times$ across MIPLIB instances.

Although the impact of moving strong branching to the GPU has been presented only on a few MIP instances and within a solver that is admittedly less developed than commercial ones like Gurobi, the integration of this GPU mechanism within a state-of-the-art solver is a research question that deserves dedicated work beyond the scope of this paper. Initial tests with `cuOpt` 26.02 on a family of `qap` instances (see `qap10` in Table 3), in which we run the entire branch-and-bound using either BATCHLP or the default parallel simplex implementation, are promising but still too preliminary to be reported in full.

## 7. Conclusions

In this paper, we introduced a batched first-order method for solving multiple linear programs in parallel on GPUs, focusing on the MIP challenges of strong branching and optimization-based bound tightening. Our approach extends the primal-dual hybrid gradient algorithm to exploit matrix-matrix operations on modern GPU architectures. Our computational experiments demonstrate significant speedups over traditional simplex-based methods, particularly on instances with large problem sizes and many subproblems per batch. We do not claim universal acceleration: the regime where BATCHLP dominates is precisely the one characterized in Section 6, with large $|S|$, moderate-density coefficient matrices, and many simplex pivots per LP. This advancement offers the MIP community a powerful tool for handling repeated subproblem solutions more efficiently, paving the way for fully incorporating GPUs in MIP algorithms.

**Future directions.** Three directions emerge naturally from our experiments. First, a *hybrid simplex/*BATCHLP *branch-and-bound* strategy, in which BATCHLP is used at the root (where $|S|$ is large and subproblems are dissimilar) and warm-started dual simplex is used deeper in the tree (where $|S|$ shrinks and subproblems become similar to their parents). Second, *batched cut generation*: solving the cut-generating LP for each single-variable disjunction is a batched LP task with the same structure as FSB, and we have already identified it as a viable target. Third, *learning-based branching*: most learning approaches in this area were designed to avoid the cost of strong branching, but BATCHLP makes per-instance strong branching affordable, which enables instance-specific learning tasks that do not suffer from the generalization issues of approaches such as (Gasse et al., 2019; Gupta et al., 2020).

## Acknowledgments

Bartolomeo Stellato is supported by the NSF CAREER Award ECCS-2239771 and the ONR YIP Award N000142512147. Stefano Gualandi acknowledges the contribution of the National Recovery and Resilience Plan, Mission 4 Component 2-Investment 1.4-National Center for HPC, Big Data and Quantum Computing (project code: CN_00000013), funded by the European Union-NextGenerationEU. The authors are thankful for NVIDIA for providing credits to access `brev` cloud infrastructure with DGX B200 GPUs.

## Impact Statement

This paper presents work whose goal is to advance the field of mathematical optimization. Our methods improve the computational efficiency of solving mixed-integer programs by exploiting GPU parallelism, which may reduce energy consumption for large-scale optimization tasks. There are many potential societal consequences of our work, none of which we feel must be specifically highlighted here.

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

## A. FSB Runtime Results

Table 1 presents detailed runtime results for the FSB experiments discussed in Section 6.

## B. OBBT Results

Table 2 presents detailed results for the OBBT experiments discussed in Section 6.

## C. MIPLIB 2017 Results

Table 3 presents detailed results for the MIPLIB 2017 experiments discussed in Section 6.

To isolate the gain due to batching from the gain due to running PDLP on a GPU, Table 4 compares sequential PDLP and batched BATCHLP on a subset of MIPLIB instances; both run on the same GPU, so the only difference is the batching mechanism. Speedups range from $1.2\times$ on `chromaticindex512-7` to $160\times$ on `cod105`, confirming that the batching is responsible for most of the gain reported in Table 3.

## D. Combinatorial Auctions and Independent Set instances

Table 5 shows the computational comparison between GUROBI and BATCHLP over combinatorial auctions and maximum independent set problems from (Gasse et al., 2019). The 'ratio' column shows that for most random maximum-independent-set instances, BATCHLP obtains moderate average speedups, except for two cases where dual simplex matches or beats BATCHLP; for combinatorial auctions, GUROBI is more often faster.

## E. Comparison of different computational environments

Table 7 reports the results of running the same version of the code on two different hardware architectures: a B200 on the `brev` NVIDIA cloud infrastructure mounted on a virtual machine equipped with an AMD EPYC 9575F 64-core processor (high-end server), and an RTX 4500 ADA mounted on a Dell workstation with an Intel Xeon CPU (low-end workstation).

*Table 1.* Runtime (in seconds) comparison of dual simplex and BATCHLP as implemented in `cuOpt` 26.02.

| Instance | Problem | | | | cuOpt 26.02 – Dual Simplex | | | | BATCHLP |
|---|---|---|---|---|---|---|---|---|---|
| | $m$ | $n$ | $nnz$ | $|S|$ | 8 threads | 16 threads | 32 threads | 64 threads | GPU B200 |
| **Combinatorial Auctions** | | | | | | | | | |
| inst_100_1500 | 360 | 1399 | 8418 | 154 | 0.03 | 0.02 | 0.02 | 0.03 | 0.05 |
| inst_300_1500 | 557 | 1453 | 8547 | 512 | 0.60 | 0.41 | 0.26 | 0.27 | 0.09 |
| inst_300_3000 | 857 | 2840 | 16635 | 514 | 0.73 | 0.48 | 0.31 | 0.36 | 0.08 |
| inst_500_1500 | 755 | 1466 | 7989 | 800 | 2.24 | 1.42 | 1.04 | 0.97 | 0.20 |
| inst_500_3000 | 1025 | 2867 | 16493 | 846 | 2.51 | 1.51 | 1.24 | 1.04 | 0.09 |
| | | | **Total runtime** | | **6.11** | **3.84** | **2.87** | **2.67** | **0.51** |
| | | | **Speedup** | | **1.0** | **1.6** | **2.1** | **2.3** | **12.0** |
| **Set Covering** | | | | | | | | | |
| inst_1000r_1000c_0.01 | 1000 | 1000 | 10000 | 330 | 1.15 | 0.63 | 0.47 | 0.49 | 0.11 |
| inst_1000r_1000c_0.05 | 1000 | 1000 | 50000 | 136 | 0.44 | 0.27 | 0.19 | 0.20 | 0.06 |
| inst_1000r_2000c_0.01 | 1000 | 2000 | 20000 | 309 | 1.22 | 0.72 | 0.50 | 0.51 | 0.10 |
| inst_1000r_2000c_0.05 | 1000 | 2000 | 100000 | 116 | 0.31 | 0.20 | 0.15 | 0.18 | 0.06 |
| inst_1000r_3000c_0.01 | 1000 | 3000 | 30000 | 265 | 0.77 | 0.54 | 0.37 | 0.37 | 0.09 |
| inst_1000r_3000c_0.05 | 1000 | 3000 | 150000 | 127 | 0.59 | 0.35 | 0.24 | 0.27 | 0.07 |
| inst_2000r_1000c_0.01 | 1978 | 999 | 19781 | 382 | 2.58 | 1.81 | 1.18 | 1.18 | 0.12 |
| inst_2000r_1000c_0.05 | 2000 | 1000 | 100000 | 182 | 1.96 | 1.06 | 0.79 | 0.76 | 0.07 |
| inst_2000r_2000c_0.01 | 2000 | 2000 | 40000 | 449 | 5.96 | 4.16 | 2.69 | 2.57 | 0.15 |
| inst_2000r_2000c_0.05 | 2000 | 2000 | 200000 | 170 | 1.73 | 1.00 | 0.73 | 0.76 | 0.08 |
| inst_3000r_1000c_0.01 | 3000 | 1000 | 30000 | 493 | 8.71 | 5.31 | 3.69 | 3.51 | 0.15 |
| inst_3000r_1000c_0.05 | 3000 | 1000 | 150000 | 164 | 1.67 | 0.98 | 0.77 | 0.72 | 0.07 |
| inst_3000r_3000c_0.01 | 3000 | 3000 | 90000 | 522 | 17.60 | 8.56 | 6.21 | 5.93 | 0.17 |
| inst_3000r_3000c_0.05 | 3000 | 3000 | 450000 | 191 | 4.14 | 2.25 | 1.82 | 1.78 | 0.11 |
| | | | **Total runtime** | | **48.83** | **27.84** | **19.80** | **19.23** | **1.41** |
| | | | **Speedup** | | **1.0** | **1.8** | **2.5** | **2.5** | **34.6** |
| **Maximum Independent Set** | | | | | | | | | |
| inst_1000_12 | 8060 | 1000 | 18815 | 1888 | 47.71 | 22.91 | 17.69 | 17.20 | 0.29 |
| inst_1000_4 | 8012 | 1000 | 18770 | 1882 | 62.79 | 31.74 | 24.24 | 23.38 | 0.28 |
| inst_1000_8 | 8209 | 1000 | 18982 | 1870 | 63.23 | 32.43 | 22.56 | 21.33 | 0.29 |
| inst_2000_12 | 18992 | 2000 | 41213 | 3320 | 88.56 | 52.97 | 34.86 | 35.01 | 0.87 |
| inst_2000_4 | 18832 | 2000 | 40960 | 3320 | 65.39 | 35.53 | 27.00 | 26.16 | 0.83 |
| inst_2000_8 | 18961 | 2000 | 41100 | 3328 | 77.92 | 40.23 | 29.28 | 29.09 | 0.84 |
| inst_3000_12 | 35612 | 3000 | 71338 | 5226 | 115.12 | 63.56 | 43.92 | 45.12 | 2.02 |
| inst_3000_4 | 35582 | 3000 | 71289 | 5218 | 130.66 | 71.48 | 52.26 | 52.57 | 1.99 |
| inst_3000_8 | 35604 | 3000 | 71325 | 5204 | 105.64 | 69.68 | 42.99 | 43.46 | 1.96 |
| | | | **Total runtime** | | **757.02** | **420.53** | **294.80** | **293.32** | **9.37** |
| | | | **Speedup** | | **1.0** | **1.8** | **2.6** | **2.6** | **80.8** |
| **Facility Location** | | | | | | | | | |
| inst_1200_200_10 | 241401 | 240200 | 960400 | 92 | 410.71 | 283.56 | 247.59 | 215.15 | 0.41 |
| inst_1200_200_15 | 241401 | 240200 | 960400 | 88 | 192.56 | 117.60 | 96.49 | 100.83 | 0.40 |
| inst_1200_200_5 | 241401 | 240200 | 960400 | 62 | 135.40 | 110.95 | 88.99 | 81.05 | 0.38 |
| inst_400_200_10 | 80601 | 80200 | 320400 | 40 | 22.52 | 18.09 | 18.05 | 17.97 | 0.12 |
| inst_400_200_15 | 80601 | 80200 | 320400 | 26 | 24.68 | 16.73 | 19.17 | 17.42 | 0.11 |
| inst_400_200_5 | 80601 | 80200 | 320400 | 32 | 4.84 | 4.97 | 5.64 | 5.05 | 0.11 |
| inst_800_200_10 | 161001 | 160200 | 640400 | 58 | 132.33 | 93.22 | 81.69 | 85.90 | 0.24 |
| inst_800_200_15 | 161001 | 160200 | 640400 | 60 | 108.64 | 76.74 | 69.47 | 65.15 | 0.22 |
| inst_800_200_5 | 161001 | 160200 | 640400 | 32 | 33.22 | 23.98 | 30.13 | 24.30 | 0.19 |
| | | | **Total runtime** | | **1064.90** | **745.84** | **657.22** | **612.82** | **2.18** |
| | | | **Speedup** | | **1.0** | **1.4** | **1.6** | **1.7** | **488.5** |

*Table 2.* Results for OBBT applied after MIP presolve of Gurobi. Dual simplex baseline: Gurobi 13.0 on an Intel Xeon Platinum 8592. BATCHLP runs on an NVIDIA B200 GPU with iteration limit 100 000 (always hit).

| Instance | $m$ | $n$ | den | Dual simplex | | BATCHLP | | | var changed | |
| | | | | runtime | pivots | runtime | Speedup | solved | num | perc. |
|---|---|---|---|---|---|---|---|---|---|---|
| test_120 | 1564 | 1767 | 1.8% | 507.9 | 755.9 | 20.1 | 25.3 | 3429/3534 | 141 | 4.0% |
| test_232 | 1296 | 1491 | 2.0% | 271.2 | 594.1 | 16.6 | 16.4 | 2866/2982 | 105 | 3.5% |
| test_233 | 1315 | 1512 | 1.9% | 225.8 | 495.6 | 11.9 | 19.0 | 2973/3024 | 110 | 3.6% |
| test_428 | 1485 | 1617 | 2.0% | 551.7 | 887.6 | 21.6 | 25.6 | 2859/3234 | 93 | 2.9% |
| test_438 | 1229 | 1332 | 2.3% | 263.7 | 660.7 | 12.0 | 21.9 | 2591/2664 | 112 | 4.2% |
| test_511 | 1100 | 1208 | 2.8% | 250.8 | 734.4 | 13.4 | 18.7 | 1800/2416 | 168 | 7.0% |
| test_523 | 932 | 1036 | 3.1% | 145.0 | 622.4 | 11.9 | 12.1 | 1981/2072 | 134 | 6.5% |
| test_53 | 1635 | 1741 | 1.6% | 485.3 | 760.3 | 17.1 | 28.5 | 3287/3482 | 108 | 3.1% |
| test_561 | 1457 | 1643 | 2.0% | 573.8 | 902.9 | 13.4 | 43.0 | 3167/3286 | 134 | 4.1% |
| test_584 | 883 | 980 | 3.2% | 141.2 | 666.9 | 9.3 | 15.2 | 1922/1960 | 117 | 6.0% |
| test_604 | 1362 | 1529 | 2.1% | 385.5 | 719.3 | 15.3 | 25.3 | 2730/3058 | 59 | 1.9% |
| test_633 | 1453 | 1614 | 1.9% | 462.0 | 786.2 | 13.2 | 34.9 | 3119/3228 | 150 | 4.7% |
| test_64 | 1589 | 1736 | 1.7% | 503.6 | 766.2 | 19.7 | 25.6 | 3290/3472 | 108 | 3.1% |
| test_698 | 1347 | 1494 | 2.0% | 309.5 | 665.9 | 11.5 | 26.9 | 2915/2988 | 115 | 3.9% |
| test_704 | 1574 | 1731 | 1.7% | 513.5 | 769.3 | 15.7 | 32.6 | 3369/3462 | 129 | 3.7% |
| test_728 | 1410 | 1570 | 2.0% | 393.1 | 710.2 | 13.6 | 28.9 | 3004/3140 | 121 | 3.9% |
| test_744 | 1575 | 1801 | 1.6% | 435.6 | 662.8 | 16.3 | 26.7 | 3513/3602 | 124 | 3.4% |
| test_751 | 1672 | 1927 | 1.6% | 613.6 | 784.3 | 14.1 | 43.6 | 3751/3854 | 132 | 3.4% |
| test_772 | 907 | 973 | 3.2% | 128.9 | 633.0 | 11.5 | 11.2 | 1572/1946 | 99 | 5.1% |
| test_791 | 1410 | 1548 | 2.1% | 432.0 | 773.0 | 17.1 | 25.3 | 2899/3096 | 100 | 3.2% |
| mean | | | | 379.7 | | 14.8 | 25.7 | | | 4.1% |

*Table 3.* Comparison of `cuOpt` dual simplex running on 28 threads (28 physical cores) with BATCHLP running on an NVIDIA H100 GPU.

| Instance | $m$ | $n$ | $nnz$ | $|S|$ | Dual simplex | BATCHLP | Speedup |
|---|---|---|---|---|---|---|---|
| chromaticindex512-7 | 33791 | 36864 | 135156 | 51604 | 3522.4 | 20.0 | 176.2 |
| cod105 | 1024 | 1024 | 57344 | 256 | 212.8 | 0.2 | 1251.7 |
| cvs16r128-89 | 4633 | 3472 | 12528 | 6420 | 20.9 | 1.3 | 16.6 |
| fast0507 | 482 | 62171 | 401756 | 556 | 5.8 | 0.3 | 16.9 |
| fiball | 2387 | 32899 | 101452 | 518 | 22.8 | 37.1 | 0.6 |
| glass-sc | 6119 | 214 | 63918 | 202 | 1.0 | 1.1 | 0.9 |
| hypothyroid-k1 | 5189 | 2595 | 431326 | 4806 | 3292.7 | 627.9 | 5.2 |
| neos-5052403-cygnet | 19134 | 27593 | 2448853 | 1272 | 63.7 | 35.8 | 1.8 |
| opm2-z10-s4 | 146325 | 5958 | 340288 | 11170 | 1921.5 | 44.1 | 43.6 |
| qap10 | 1820 | 4150 | 18200 | 2442 | 28.8 | 1.9 | 15.4 |
| splice1k1 | 6504 | 3252 | 1758012 | 3816 | 3355.3 | 1468.6 | 2.3 |

*Table 4.* Sequential PDLP versus batched BATCHLP on the same GPU, isolating the effect of batching. Runtimes are in seconds.

| Instance | $|S|$ | Sequential PDLP | BATCHLP | Speedup |
|---|---|---|---|---|
| chromaticindex512-7 | 16758 | 724.5 | 603.0 | 1.2 |
| cod105 | 1388 | 32.0 | 0.2 | 160.1 |
| cvs16r128-89 | 6834 | 1007.6 | 20.7 | 48.7 |
| fast0507 | 582 | 340.7 | 50.3 | 6.8 |
| fiball | 778 | 54.7 | 6.3 | 8.7 |
| glass-sc | 202 | 74.5 | 5.0 | 14.9 |
| qap10 | 2658 | 167.6 | 1.8 | 91.1 |

*Table 5.* Results on random instances generated as in (Gasse et al., 2019): combinatorial auctions, maximum independent set of small or medium size.

| | | | GUROBI | | | | BATCHLP | | |
|---|---|---|---|---|---|---|---|---|---|
| co. auctions | item | bids | root | total | pivots | $|S|$ | time | iter | ratio |
| | 100 | 1500 | 0.0 | 0.2 | 28.9 | 154 | 0.4 | 4000 | 0.4 |
| | 300 | 1500 | 0.1 | 3.8 | 172.7 | 512 | 0.4 | 3000 | 8.7 |
| | 300 | 3000 | 0.1 | 3.5 | 142.1 | 514 | 0.5 | 4000 | 7.1 |
| | 500 | 1500 | 0.1 | 10.9 | 264.3 | 800 | 0.7 | 3000 | 16.5 |
| | 500 | 3000 | 0.2 | 14.2 | 267.9 | 846 | 0.6 | 3000 | 25.4 |
| ind. set | nodes | affinity | | | | | | | |
| | 1000 | 12 | 0.1 | 6.8 | 94.6 | 1824 | 2.0 | 2000 | 3.4 |
| | 1000 | 4 | 0.1 | 6.2 | 88.5 | 1850 | 1.7 | 2000 | 3.7 |
| | 1000 | 8 | 0.1 | 6.2 | 86.6 | 1862 | 1.7 | 2000 | 3.7 |
| | 2000 | 12 | 0.2 | 22.8 | 112.0 | 3772 | 9.0 | 2000 | 2.5 |
| | 2000 | 4 | 0.2 | 21.7 | 108.9 | 3782 | 21.4 | 2000 | 1.0 |
| | 2000 | 8 | 0.2 | 21.9 | 111.4 | 3792 | 24.9 | 2000 | 0.9 |
| | 3000 | 12 | 0.5 | 46.0 | 125.6 | 5772 | 27.4 | 2000 | 1.7 |
| | 3000 | 4 | 0.4 | 46.2 | 125.6 | 5750 | 27.1 | 2000 | 1.7 |
| | 3000 | 8 | 0.4 | 46.5 | 125.8 | 5766 | 27.1 | 2000 | 1.7 |

*Table 6.* Results on ORLIB instances comparing the speedups obtained with high vs. low accuracies.

| | | | | | | | | | BATCHLP | | | | |
|---|---|---|---|---|---|---|---|---|---|---|---|---|---|
| | | | | | | GUROBI | | $\epsilon = 10^{-4}$ | | | $\epsilon = 10^{-8}$ | | |
| | $m$ | $n$ | nnz | den. | $|S|$ | time | time | it. | ratio | time | it. | ratio |
| scpcyc06 | 240 | 192 | 960 | 2.1% | 120 | 0.4 | 0.0 | 140 | 7.8 | 0.1 | 320 | 6.5 |
| scpcyc07 | 672 | 448 | 2688 | 0.9% | 224 | 2.5 | 0.1 | 160 | 36.2 | 0.1 | 370 | 25.7 |
| scpcyc08 | 1792 | 1024 | 7168 | 0.4% | 786 | 67.3 | 0.3 | 230 | 193.3 | 0.6 | 500 | 107.0 |
| scpcyc09 | 4608 | 2304 | 18432 | 0.2% | 1783 | 1559.8 | 1.7 | 300 | 909.5 | 5.0 | 810 | 311.8 |
| scpcyc10 | 11520 | 5120 | 46080 | 0.1% | 2944 | 27937.9 | 7.8 | 340 | 3587.3 | 14.9 | 630 | 1876.8 |
| scpclr10 | 511 | 210 | 13230 | 12.3% | 135 | 0.8 | 0.1 | 430 | 8.5 | 0.2 | 1170 | 4.4 |
| scpclr11 | 1023 | 330 | 41910 | 12.4% | 330 | 37.9 | 0.3 | 670 | 111.5 | 1.4 | 2630 | 27.5 |
| scpclr12 | 2047 | 495 | 126225 | 12.5% | 342 | 138.4 | 1.0 | 750 | 138.4 | 2.2 | 1660 | 61.9 |
| scpclr13 | 4095 | 715 | 365365 | 12.5% | 715 | 2246.4 | 10.8 | 1730 | 207.2 | 57.8 | 7390 | 38.9 |

*Table 7.* Comparison on random set covering instances generated as in (Gasse et al., 2019), with 2000 constraints, different number of variables $n$ and maximum coefficient $c_i$ between (1) low-end workstation, and (2) high-end server. $CPU$ and $GPU$ columns refer to GUROBI dual simplex and BATCHLP, respectively. The instances of density 1% with 10 000 variables are unsolved on the low-end workstation due to memory limits.

| | | | | | | Low-End Workstation | | | High-end Server | | | |
|---|---|---|---|---|---|---|---|---|---|---|---|---|
| | | | | | | runtime | | ratio | runtime | | ratio | |
| den. | $c_i$ | $n$ | $\lvert S\rvert$ | Pivots | Iter | $CPU_1$ | $GPU_1$ | $\frac{CPU_1}{GPU_1}$ | $CPU_2$ | $GPU_2$ | $\frac{CPU_2}{GPU_2}$ | $\frac{GPU_1}{GPU_2}$ |
| 0.5% | 2 | 10000 | 3018 | 1589.4 | 6000 | 9029.5 | 52.7 | 171.4 | 6147.3 | 4.5 | 1355.9 | 11.6 |
| | | 8000 | 2836 | 1494.9 | 5000 | 6093.3 | 38.0 | 160.3 | 4204.7 | 3.5 | 1190.9 | 10.8 |
| | | 6000 | 2654 | 1280.4 | 6000 | 3576.5 | 64.3 | 55.6 | 2517.2 | 2.6 | 963.7 | 24.6 |
| | | 4000 | 2214 | 1014.3 | 5000 | 1281.7 | 16.4 | 78.0 | 880.9 | 1.8 | 494.2 | 9.2 |
| | 5 | 10000 | 2314 | 1059.3 | 8000 | 1706.7 | 54.9 | 31.1 | 1177.4 | 4.7 | 247.9 | 11.6 |
| | | 8000 | 2144 | 961.8 | 8000 | 1159.3 | 34.0 | 34.1 | 806.2 | 4.4 | 181.6 | 7.7 |
| | | 6000 | 1930 | 777.8 | 8000 | 649.9 | 26.4 | 24.6 | 462.2 | 2.7 | 172.7 | 9.9 |
| | | 4000 | 1740 | 669.4 | 9000 | 298.0 | 18.9 | 15.8 | 218.7 | 2.2 | 98.4 | 8.5 |
| | 10 | 10000 | 1804 | 741.2 | 12000 | 493.8 | 52.2 | 9.5 | 351.5 | 5.2 | 67.5 | 10.0 |
| | | 8000 | 1790 | 727.1 | 11000 | 415.0 | 43.2 | 9.6 | 317.9 | 4.2 | 75.9 | 10.3 |
| | | 6000 | 1650 | 601.7 | 13000 | 223.6 | 30.6 | 7.3 | 165.6 | 3.3 | 49.7 | 9.2 |
| | | 4000 | 1552 | 519.6 | 12000 | 141.1 | 18.2 | 7.8 | 103.2 | 2.9 | 35.2 | 6.2 |
| 1.0% | 2 | 10000 | 2898 | 2194.7 | 6000 | - | - | - | 15550.6 | 6.3 | 2479.4 | - |
| | | 8000 | 2754 | 1897.8 | 6000 | 14177.3 | 55.1 | 257.5 | 9912.6 | 4.8 | 2069.7 | 11.5 |
| | | 6000 | 2400 | 1635.6 | 6000 | 6991.6 | 32.8 | 213.4 | 4714.0 | 4.1 | 1163.9 | 8.1 |
| | | 4000 | 2096 | 1305.6 | 6000 | 3071.8 | 18.9 | 162.6 | 2154.0 | 2.7 | 811.3 | 7.1 |
| | 5 | 10000 | 2116 | 1287.5 | 9000 | - | - | - | 2292.0 | 6.7 | 339.6 | - |
| | | 8000 | 1960 | 1165.7 | 9000 | 2164.2 | 51.8 | 41.8 | 1467.1 | 5.4 | 273.7 | 9.7 |
| | | 6000 | 1652 | 948.4 | 10000 | 919.1 | 35.0 | 26.3 | 615.3 | 3.6 | 171.3 | 9.7 |
| | | 4000 | 1488 | 786.5 | 10000 | 521.9 | 20.3 | 25.7 | 358.0 | 2.9 | 122.6 | 7.0 |
| | 10 | 10000 | 1570 | 843.5 | 16000 | - | - | - | 519.0 | 7.8 | 66.8 | - |
| | | 8000 | 1420 | 778.4 | 15000 | 513.6 | 56.1 | 9.2 | 348.9 | 5.6 | 62.0 | 10.0 |
| | | 6000 | 1338 | 732.5 | 14000 | 361.7 | 40.7 | 8.9 | 245.2 | 13.3 | 18.4 | 3.1 |
| | | 4000 | 1196 | 539.4 | 16000 | 156.6 | 27.0 | 5.8 | 110.2 | 3.2 | 34.6 | 8.5 |

