# OpenReview forum: "Batched First-Order Methods for Parallel LP Solving in MIP"
_ICML.cc/2026/Conference — ICML 2026 regular_

### Official Review · Reviewer_uBJP · 2026-02-25

**Soundness:** 3
**Presentation:** 4
**Significance:** 2
**Originality:** 2
**Overall Recommendation:** 4
**Confidence:** 4

**Summary:**

The paper introduces a batched FOM (PDLP) for GPU acceleration to solve mixed-integer programs (MIPs). Specificallym it integrates key PDLP components, such as restarts and adaptive primal weight/step-size adjustments, into a batched GPU implementation. The proposed framework is subsequently embedded within a Branch-and-Bound procedure to solve the LP relaxations at each node. Numerical experiments are presented.

**Compliance With Llm Reviewing Policy:**

Affirmed.

**Final Justification:**

The authors have properly resolved all of my concerns, and I decide to raise my score.

**Key Questions For Authors:**

I personally find the core idea of the paper interesting, but several crucial problems in the experimental section need to be addressed:

1. In Table 1, the proposed framework is evaluated on GPUs, while the baseline cuOPT is run on CPU. This comparison seems unfair. CuOPT is specifically designed and optimized for CUDA/GPU execution, and the chosen CPU is a workstation-grade processor, typically weaker in per-core performance, and CPU core counts are far lower than GPU streaming multiprocessors. The authors should clearly justify the hardware choices and explain why a GPU vs. CPU comparison is meaningful in this context.
2. COPT has released an implementation of PDLP (see https://github.com/COPT-Public/cuPDLP-C). Including a comparison against it would significantly strengthen the empirical evaluation and better demonstrate the advantages (or limitations) of the proposed batched approach.
3. The discussion of Table 3 is absent. It is interesting that the proposed framework slows down the solution process on several instances, especially those are not the smallest ones. Could this be caused by numerical issues, such as ill-conditioned matrices, near-degenerate constraints, or binding constraints causing instability FOMs? A detailed analysis would be valuable.

**Limitations:**

yes

**Strengths And Weaknesses:**

Strength:
1. The presentation is clear, well-organized, and easy to follow.
2. The idea of OBBT is novel and interesting, offering a fresh perspective on accelerating the MIP solver using GPUs.

Weakness:
1. The comparison experiments are not entirely fair with respect to hardware specifications.
2. The core of the proposed framework appears to be a relatively simple modification of the original PDLP algorithm [1].

[1] Applegate, David, et al. "Pdlp: A practical first-order method for large-scale linear programming." arXiv preprint arXiv:2501.07018 (2025).

---

> ### Author Rebuttal · Authors · 2026-03-31
>
> We thank the reviewers for the comment about the **relatively simple modification of the original PDLP algorithm** because it highlights a clear flaw of the paper that failed describing the technical challenges of developing the batched algorithm.
>
> **W2: Core of framework is a relatively simple modification of PDLP**
>
> The change from sequential to batched PDHG of the main iterations is relatively intuitive: replace each SpMV (matrix-vector) operation with SpMM (matrix-matrix) operations, since the main steps $A x$ and $A^\top y$ become $A X$ and $A^\top Y$, where $A$ is a sparse matrix and $X$ and $Y$ are dense matrices. For the main steps of the algorithm (iterated projections), matrices $X$ and $Y$ are stored in row-major format. For the projection operators and the other updates, we have new ad-hoc CUDA kernels, since cuBLAS is no longer sufficient by itself.
>
> The challenges lie in the details of the algorithm:
>
> *Periodic checks* (once every $k$ PDHG iterations) of primal feasibility, dual feasibility, and optimality tolerances have to be carried out for each column of matrices $X$ and $Y$. This is the first point where we switch from row-major to column-major format. The same holds for infeasibility detection.
>
> *Early termination*: if any subproblem meets the stopping criteria, we have to extract the corresponding primal and dual solutions (column vectors) from matrices $X$ and $Y$. This is faster if we convert from row-major to column-major, remove the solved columns, and convert back to row-major. These operations also require new ad-hoc CUDA kernels.
>
> *Restart strategy*: the restart criteria are more complex for batched PDLP because we want to restart the entire batch simultaneously. We therefore derive a restarting condition based on an average restart criterion across subproblems — as a consequence, some subproblems are restarted too early and others too late.
>
> *Warm restart*: handling warm restarts for each subproblem after the solution of the initial root LP is non-trivial, since every subproblem may have different warming conditions.
>
> *Removing efficiently*: the removal is also not trivial for the vector slices in the dense matrices. We cannot just create a hole in the matrix so we have to swap a non-converging slice from the end and swap it at the hole position. Doing that one by one would be terribly inefficient so we first need to identify the minimum number of swaps we need to do and execute all of them in a batch manner for all vectors and matrices.
>
> *Optimal batch size*: there is the issue of computing, at runtime and per instance, the optimal batch size to best exploit GPU capabilities — this involves a tradeoff between memory usage and the speed of basic operations. While the two matrix-matrix operations are expensive, on larger instances updating the dense matrices $X$ and $Y$ also take significant time, making the choice of batch size crucial for runtime efficiency.
>
> **W1: Hardware fairness / Q1: Table 1 hardware justification**
>
> Concerning the fairness of the computational evaluations, we would like to clarify that cuOpt's dual simplex runs on *CPU*, not GPU. The name "cuOpt" may suggest GPU-only, but the dual simplex component is a standard CPU implementation. The comparison is therefore: cuOpt CPU dual simplex (multi-threaded) vs. cuOpt GPU BatchLP—same solver framework, same preprocessing, different LP backends. We will clarify this in the paper.
> The GPU-vs-CPU comparison reflects the practical deployment scenario: MIP solvers run on CPUs today; we propose *adding* a GPU for specific subroutines. Our hardware choices represent high-end configurations for both: the Grace Hopper and DGX B200 systems pair top-tier CPUs with top-tier GPUs. We also validated on a lower-end RTX 4500 workstation with consistent conclusions. CPU scaling saturates: Table 1 shows 8-to-64 thread scaling yields only 2.5×, far from the ideal 8×.
>
> **Q2: Comparison with cuPDLP-C**
>
> The above challenges are precisely what distinguishes BatchLP from running sequential PDLP on a GPU. To isolate the batching effect, we refer to our response to reviewer FNmp, comparing sequential vs. batched PDLP on 8 MIPLIB instances with speedups of 1.1×–42.4×. Hans Mittelmann's benchmarks compare cuOpt (batch size = 1) to cuPDLP and COPT: https://plato.asu.edu/ftp/lpfeas.html
>
> **Q3: Table 3 slowdowns — numerical issues?**
>
> The two slower instances have *structural* (not numerical) explanations. **fiball** (0.6×): wide sparse matrix ($n/m=13.8$), only 518 subproblems—SpMM is memory-bound, limited batching benefit. **glass-sc** (0.9× vs. 28-thread cuOpt): tall dense matrix ($m/n=28.6$), only 202 subproblems—favors direct methods. Both instances solve correctly with both methods. Notably, glass-sc achieves 9.4× speedup against *sequential* Gurobi; the apparent slowdown is entirely due to CPU parallelism closing the gap. These cases help characterize *when* GPU offloading is most beneficial: large $|S|$ and moderate-density matrices.

---

> > ### Author Rebuttal · Reviewer_uBJP · 2026-04-02
> >
> > The authors have properly resolved all of my concerns, and I decide to raise my score.

---

### Official Review · Reviewer_FNmp · 2026-02-26

**Soundness:** 3
**Presentation:** 2
**Significance:** 3
**Originality:** 2
**Overall Recommendation:** 4
**Confidence:** 2

**Summary:**

The paper introduces BatchLP, a batched first-order method designed to solve multiple related linear programs simultaneously on GPU architectures. The authors evaluate the method on two mixed-integer programming tasks: Full Strong Branching and Optimization-Based Bound Tightening. Experimental results show that BATCHLP achieves substantial speedups.

**Compliance With Llm Reviewing Policy:**

Affirmed.

**Final Justification:**

I think that accelerating optimization solvers via CUDA is a meaningful and promising research direction. Given the technical implementation and the experimental validation, this maniscript provides a valuable contribution to the field of GPU-accelerated optimization. Therefore, I maintain my positive recommendation.

**Key Questions For Authors:**

- how exactly was Gurobi configured (version, specific parameters)? Specifically, did the Gurobi baseline utilize any GPU acceleration or multi-threading?
- Since BatchLP's primary advantage is the "batched" matrix-matrix operation, a comparison with a solver that solves the same problems sequentially on a GPU would better isolate the benefits of your batched approach.

**Limitations:**

yes

**Strengths And Weaknesses:**

Strengths:
- The implementation details are efficient, utilizing modern libraries like cuSPARSE and cuBLAS and avoiding dynamic memory allocation at runtime to ensure performance.
- The reported speedups are impressive.

Weaknesses:
- The evaluation compares BATCHLP against cuOpt's dual simplex and Gurobi. However, it lacks a comparison with existing GPU implementations of PDHG (such as cuPDLP.jl or the GPU version of PDLP mentioned in the literature review ). It is unclear how much of the gain is due to the "batched" matrix-matrix formulation versus standard GPU-based PDHG acceleration.
- The paper mentions several hyperparameters. However, there is no detailed explanation of how these were tuned or how sensitive the performance is to these choices.

---

> ### Author Rebuttal · Authors · 2026-03-31
>
> We thank the reviewer for the insightful feedback that highlights an important discussion that we missed, the **effect of batching LPs vs solving them on GPU**. That is the core contribution of the paper and in order to isolate the impact of the batching mechanism we report the following table:
>
> **W1: Lacks comparison with GPU PDHG / Q2: Sequential GPU comparison to isolate batching benefits**
>
> | Instance | \|S\| | Sequential PDLP | Batched PDLP | Speedup |
> |---|---|---|---|---|
> | chromaticindex512-7 | 16758 | 724.5 | 603.0 | 1.2 |
> | cod105 | 1388 | 32.0 | 0.2 | 160.1 |
> | cvs16r128-89 | 6834 | 1007.6 | 20.7 | 48.7 |
> | fast0507 | 582 | 340.7 | 50.3 | 6.8 |
> | fiball | 778 | 54.7 | 6.3 | 8.7 |
> | glass-sc | 202 | 74.5 | 5.0 | 14.9 |
> | qap10 | 2658 | 167.6 | 1.8 | 91.1 |
>
> Precisely, the table reports the computing times and speedup on 7 instances of the MIPlib (included from Table 3 of the submission), where the batching advantage over sequential PDLP on the GPU is clear.
>
> **W2: Hyperparameter sensitivity not addressed**
>
> Most hyperparameters follow standard cuPDLPx defaults (Lu & Yang 2024) and are not tuned per-instance. Tolerance sensitivity: BatchLP remains faster at both eps=1e-4 and eps=1e-8. For FSB, eps=1e-4 suffices because branching scores are used as heuristic ordering; for OBBT, eps_dual=1e-8 for safety (Sec 6.2) since bounds must be valid.
>
> Some hyperparameters are associated with the modification required to change from sequential to batched PDHG of the main iterations. Such a change is relatively intuitive: replace each SpMV (matrix-vector) operation with SpMM (matrix-matrix) operations, since the main steps $A x$ and $A^\top y$ become $A X$ and $A^\top Y$, where $A$ is a sparse matrix and $X$, $Y$ are dense matrices. For the main steps of the algorithm (iterated projections), dense matrices $X$ and $Y$ are stored in row-major format. For the projection operators and the other updates, we have new ad-hoc CUDA kernels, since cuBLAS is no longer sufficient by itself.
>
> The challenges lie in the details of the algorithm:
>
> *Periodic checks* (once every k PDHG iterations) of primal feasibility, dual feasibility, and optimality tolerances have to be carried out for each column of matrices $X$ and $Y$. This is the first point where we switch $X$ and $Y$ from row-major to column-major format. The same holds for infeasibility detection.
>
> *Early termination*: if any subproblem meets the stopping criteria, we have to extract the corresponding primal and dual solutions (column vectors) from matrices $X$ and $Y$. This is faster if we convert from row major to column major, remove the solved columns, and convert back to row major. These operations also require new ad-hoc CUDA kernels.
>
> *Restart strategy*: the restart criteria are more complex for batched PDLP because we want to restart the entire batch simultaneously. We therefore derive a restarting condition based on an average restart criterion across subproblems — as a consequence, some subproblems are restarted too early and others too late.
>
> *Warm restart*: handling warm restarts for each subproblem after the solution of the initial root LP is non-trivial, since every subproblem may have different warming conditions.
>
> *Removing efficiently*: the removal is also not trivial for the vector slices in the dense matrices. We cannot just create a hole in the matrix so we have to swap a non-converging slice from the end and swap it at the hole position. Doing that one by one would be terribly inefficient so we first need to identify the minimum number of swaps we need to do and execute all of them in a batch manner for all vectors and matrices.
>
> *Optimal batch size*: there is the issue of computing, at runtime and per instance, the optimal batch size to best exploit GPU capabilities — this involves a tradeoff between memory usage and the speed of basic operations. While the two matrix-matrix operations are expensive, on larger instances updating the dense matrices $X$ and $Y$ also take significant time, making the choice of batch size crucial for runtime efficiency.
>
> **Q1: Gurobi configuration**
>
> We would like to clarify that we used Gurobi 13.0 on multi-threaded CPUs.
>
> OBBT experiments (Table 2): Gurobi 13.0, dual simplex method (Method=1), LP presolve disabled (Presolve=0), single-threaded (Threads=1). No GPU acceleration. Each of the 2n OBBT subproblems solved sequentially.
>
> FSB experiments (Table 1): cuOpt 26.02's dual simplex implementation (not Gurobi) with multi-threading (8, 16, 32, 64 threads). Each thread solves subproblems from its share of |S| sequentially using warm-started dual simplex from the root basis.
>
> MIPLIB experiments (Table 3): cuOpt 26.02's dual simplex on 28 threads (28 physical cores), with a 200-pivot limit per subproblem (the default work limit in cuOpt).

---

> > ### Author Rebuttal · Reviewer_FNmp · 2026-04-04
> >
> > Thanks for your response. I will keep my score for acceptance.

---

### Official Review · Reviewer_Bp66 · 2026-03-09

**Soundness:** 3
**Presentation:** 2
**Significance:** 3
**Originality:** 3
**Overall Recommendation:** 3
**Confidence:** 3

**Summary:**

This article proposes BATCHLP, a GPU-accelerated batch processing first-order method for solving the batch processing LP subproblem in FSB and OBBT tasks for MIP. By extending the PDHG algorithm, this method replaces matrix-vector operations with GPU-optimized matrix-matrix operations, thereby solving the problems of high data transmission costs and insufficient parallelism in CPU-GPU architectures. Experiments on multiple benchmark datasets have shown that this method achieves significant acceleration compared to traditional simplex methods and the Gurobi solver.

**Compliance With Llm Reviewing Policy:**

Affirmed.

**Final Justification:**

The author has addressed all concerns. Currently, the charts in the article, such as bar graphs, are presented in a black color scheme, which impacts the readability. If the chart styles are adjusted as the author acknowledges is possible, there would be a significant improvement in the overall presentation of the article.

**Key Questions For Authors:**

1. The paper reports impressive speedups for specific subroutines like FSB and OBBT. However, these subroutines often constitute only a fraction of the total time in a complete Branch-and-Bound (B&B) tree search, especially on hard MIPLIB instances where node processing overhead and other heuristics dominate. Can you provide experimental results showing the total solution times for a representative set of MIPLIB 2017 instances solved to optimality (or a fixed gap), comparing your GPU-accelerated solver against the CPU-only baseline? Demonstrating significant total time reduction would strongly validate the practical significance of the work.
2. How does BATCHLP’s performance compare to Warm-Start Dual Simplex specifically in the deeper levels of the B&B tree?
The introduction acknowledges that the Simplex method's warm start is strong, while first-order methods typically have slower cold starts. How can the authors compensate for this disadvantage in batch settings?
3. Does the convergence of the batch PDHG algorithm depend on subproblem consistency? Furthermore, for LP subproblems with slight variable bound differences in strong branching and OBBT tasks, is there a more targeted convergence acceleration method than general batch strategies?
4. Can the method proposed in this paper be combined with currently popular learning-based branch variable selection algorithms (e.g., Gasse et al., Gupta et al.) to achieve more efficient MIP solving?

**Limitations:**

The batch PDHG method lacks independent convergence theoretical proof. There is no quantitative comparison with mainstream GPU LP solvers. The applicable scope and improvement directions should be clearly defined.

**Strengths And Weaknesses:**

Strengths:
1. The authors effectively position their work relative to two distinct lines of prior literature, clearly distinguishing their "accelerated exact computation" approach from "learning-based approximation" methods (e.g., Gasse et al., Gupta et al.).
2. The experiments are well-designed and cover diverse scenarios. For example, the evaluation includes comparisons against strong, industry-standard baselines (e.g., multi-threaded CPU dual simplex in cuOpt/Gurobi). Furthermore, the paper reports not only speedup metrics but also rigorously verifies the solution quality, ensuring the reliability of the accelerated results.
3. This work effectively improves the efficiency of MIP batch LP solving and provides a new paradigm for GPU acceleration in MIP solvers.

Weaknesses:
1. The paper demonstrates significant acceleration for individual steps (FSB/OBBT) but lacks a comparison of end-to-end MIP solution times. This is a key metric for evaluating the value of MIP algorithms in the optimization field; the absence of this data means that some conclusions in the paper lack direct evidentiary support.
2. This method appears effective only on specific types of problems (large-scale, high parallelism) and performs poorly on some instances from standard benchmark suites (e.g., MIPLIB).
3. Additionally, the experimental results lack a sensitivity analysis across different hardware configurations, and the proposed algorithm lacks a theoretical convergence analysis.
4. Although the technical depth is impressive, the current narrative structure occasionally obscures the core insights of the paper, making it difficult for readers to grasp the key points. Some charts are not aesthetically pleasing or professional enough, and some references have inconsistent formats.

---

> ### Author Rebuttal · Authors · 2026-03-31
>
> We thank the reviewer for their insightful feedback. The main issue raised is that of **end-to-end integration of batchLP within B&B**, common to other reviews.
>
> **W1: Lacks end-to-end MIP solution times / Q1: Total solution times on MIPLIB 2017**
>
> We appreciate this request and provide preliminary results (extending Section 6.3). However, integrating ideas that significantly affect B&B is inherently difficult and constitutes a research question in itself. Focusing on strong branching (SB), our contribution shows that what was previously infeasible—a full SB pass without restricting candidates or limiting pivots—is now largely achievable on GPUs. While batching LPs on GPUs is most beneficial when the number of simplex pivots is large (e.g., set covering), full B&B integration requires studying: (a) SB frequency, (b) LP precision effects on branching decisions, and (c) SB score–pseudocost interactions. This must be compared with CPU implementations where cores diverted to SB raise synchronization issues.
> Beyond experimentation, this calls for new algorithmic ideas within a CPU–GPU B&B framework that does not yet exist; even cuOpt raises challenges such as determinism. Thus, we believe that requiring full integration at this stage is premature.
> Nevertheless, we extended Section 6.3 by running cuOpt B&B after SB at the root, using GPU (batchLP) or a 64-core CPU with pivot limits (200, 500, 1000) on selected MIPLIB instances. While not conclusive, results show: (a) On qap10, large SB speedups (3.0×, 7.2×, 14.6× vs. 1.87s baseline) translate into B&B speedups (1.6×, 1.9×, 2.3× vs. 711s baseline); and (b) On neos-1582420, istanbul-no-cutoff, and swath3, SB times are comparable, but batchLP is faster overall, and more accurate SB (higher pivot limits) improves solve times.
> These experiments use a deterministic (and slower) cuOpt version to isolate SB effects by disabling GPU heuristics. Results indicate nontrivial gains in both speed (a) and accuracy (b). On many instances unsolved within 10 minutes using nondeterministic cuOpt, batchLP achieves smaller final gaps. Overall, we consider these results promising.
>
> **W2: Method effective only on specific problem types**
>
> The goal of this work is to showcase how batched first-order LP solves achieve great performance for GPU offloading in MIP. We do not claim universal acceleration, but our benchmarks—including new ones—are competitive on a wide range of instances.
>
> **W3: Hardware sensitivity and convergence / Q3: Subproblem consistency**
>
> *hardware:* We tested on H100 and B200 GPUs; Table 2 compares 8–64 CPU threads. More comparisons in the camera-ready.
> *convergence:* Halpern PDHG with restarts converges linearly (Lu & Yang, 2024). In our batched version, each column of $Z^k = (X^k,Y^k)$ evolves independently with its own step sizes. Batch coupling is only through the averaged restart criterion, benign for FSB/OBBT where subproblems share A. For inconsistent batches, convergence is preserved but restart timing may be suboptimal. Per-column adaptive step sizes (Section 4.3) and per-column restart strategies are promising extensions.
>
> **W4: Presentation and narrative**
>
> Concerning the narrative, we agree that the paper fails in providing a clear picture of the **complexity of the modifications required to effectively implement batchLP**, which are an important ingredient of our work and justify this research even without end-to-end integration. The transition from sequential to batched PDHG is conceptually simple: replace SpMV with SpMM so that $A x$ and $A^\top y$ become $A X$ and $A^\top Y$, with $X$ and $Y$ dense. Main iterations use $A$ and $A^\top$ in CSR format, while projections and updates require custom CUDA kernels beyond cuBLAS.
> The challenges arise in the details: per-column feasibility/optimality checks requiring row/column conversions; efficient early termination and column removal via custom kernels; batch-level restart strategies with averaged criteria that may restart some subproblems too early or late; complex warm restarts since subproblems differ; and batch size selection balancing memory and performance. See our answer to reviewer uBJP for a detailed description.
> We will also improve charts and fix reference formatting.
>
> **Q2: BatchLP vs warm-start dual simplex in deeper B&B levels**
>
> We warm-start from the parent LP solution. Deeper in B&B, the number of subproblems |S| decreases but subproblems are more similar. A hybrid strategy—BatchLP at root, simplex deeper—is natural.
>
> **Q4: Combination with learning-based branching (Gasse et al., Gupta et al.)**
>
> Learning-based branching methods were designed to avoid SB because of its expensiveness. With batchLP, SB becomes affordable, so in principle these approximations are not required anymore. However, batchLP allows to do SB on the fly, i.e., on each instance, which enables new learning tasks that do not suffer from generalization issues. It is indeed a beautiful research direction.

---

> > ### Author Rebuttal · Reviewer_Bp66 · 2026-04-02
> >
> > Thanks for the rebuttal. However, despite their replies to individual points, my concerns persist, particularly regarding the details of the figures and tables.

---

> > > ### Author Response · Authors · 2026-04-02
> > >
> > > Thank you for the acknowledgement. We would appreciate it if you could specify which figures and tables you have concerns about, so that we can address them precisely. The paper contains 4 figures and 9 tables spanning different experiments (FSB, OBBT, MIPLIB, hardware comparison), and understanding which specific aspects concern you would help us provide a more targeted response.
> > >
> > > In the meantime, we have already identified and will fix the following in the camera-ready version:
> > > - Improved bar charts (Figures 2–4) with distinct colors instead of grayscale shading
> > > - Consistent LaTeX table formatting (booktabs rules, column spacing)
> > > - Minor typos in table descriptions
> > >
> > > If your concern relates to the content of the tables (e.g., missing data or the end-to-end B&B results discussed in W1), we are happy to discuss further.

---

### Official Review · Reviewer_yaQM · 2026-03-13

**Soundness:** 2
**Presentation:** 2
**Significance:** 2
**Originality:** 2
**Overall Recommendation:** 4
**Confidence:** 3

**Summary:**

The paper shows how to exploit GPU parallelism to solve batches of closely related linear programs efficiently within mixed integer programming (MIP), focusing on core subtasks in MIP solving such as strong branching and optimization based bound tightening (OBBT(. The paper proposes BATCHLP, a GPU-oriented batched first-order solver that extends the primal–dual hybrid gradient (PDHG) algorithm. Instead of solving each LP independently with matrix–vector operations, the method reformulates batched updates using matrix–matrix operations, which are well suited to modern GPU architectures. The experimental results demonstrate significant speedups compared to standard multi-threaded CPU based solvers.

**Compliance With Llm Reviewing Policy:**

Affirmed.

**Final Justification:**

The rebuttal has addressed my concerns.

**Key Questions For Authors:**

1. How does using BATCHLP for strong branching affect total solve time and node counts in full branch-and-bound runs on large benchmark sets?

2. Could the same batched approach be applied to other repeated LP tasks in MIP, such as cut generation or Lagrangian relaxations?

**Limitations:**

The paper discusses its limitations

**Strengths And Weaknesses:**

Strengths:
------------

The paper is well motivated and proposes a practical solution for efficient MIP solving in practice.

The experimental evaluation is sound. The results on strong branching and OBBT show impressive speedups over both parallel CPU simplex and commercial solvers, especially for large batches.


Weaknesses:
--------------
Most experiments isolate strong branching or OBBT at the root node. Therefore, the impact on full branch-and-bound performance remains preliminary.

Although the solver is carefully engineered, integrating such GPU-based routines into mature commercial MIP solvers is nontrivial and not fully addressed.

GPU enabled heuristic search has already been investigated in the past (e.g., GPU-based A*), however the paper does not mention anything about this line of work.

---

> ### Author Rebuttal · Authors · 2026-03-31
>
> We thank the reviewer for their insightful feedback. The main issue raised by the reviewer is that of the **end-to-end integration of batchLP within branch-and-bound (B&B)**. It is common to other reviews and will be reported again later.
>
> **W1: Impact on full B&B performance remains preliminary / W2: Integration into commercial solvers / Q1: Full B&B solve time and node counts**
>
> We appreciate this request and provide preliminary results (extending Section 6.3). However, the integration of ideas that significantly affect B&B is inherently difficult to assess and constitutes a research question in itself. Focusing on strong branching (SB), our contribution shows that what was previously infeasible—performing a full SB pass without restricting candidate variables or limiting LP pivots—is now largely achievable using GPUs effectively (see next point). While batching LPs on GPUs is not free and is most beneficial when the number of Simplex pivots is large (e.g., set covering), full integration into B&B requires studying: (a) SB frequency, (b) the effect of LP precision on branching decisions, and (c) interactions between SB scores and pseudocosts. This must be compared with classical CPU implementations, where cores are diverted to SB in parallel, raising synchronization issues.
> Beyond experimentation, this calls for new algorithmic ideas within a CPU–GPU B&B framework that does not yet exist; even using cuOpt raises challenges such as determinism. Thus, we believe that requiring full integration at this stage is premature.
> Nevertheless, we extended the experiment from Section 6.3 by running cuOpt B&B after SB at the root, using either GPU (batchLP) or a 64-core CPU with pivot limits (200, 500, 1000) on selected MIPLIB instances. While not conclusive, results show: (a) On qap10, large SB speedups (3.0×, 7.2×, 14.6× vs. 1.87s baseline) translate into overall B&B speedups (1.6×, 1.9×, 2.3× vs. 711s baseline); and (b) On instances such as neos-1582420, istanbul-no-cutoff, and swath3, SB times are comparable, but batchLP is faster overall, and more accurate SB (higher pivot limits) improves solve times.
> These experiments use a deterministic (and slower) cuOpt version to isolate SB effects by disabling GPU heuristics. Results indicate that batching LPs yields nontrivial gains in both speed (a) and accuracy (b). Finally, on many instances unsolved within 10 minutes using nondeterministic cuOpt (required to consistently find primal solutions), batchLP achieves smaller final gaps. Overall, we consider these results promising.
>
> Although not explicitly raised by this reviewer, the **complexity of the modifications required to effectively implementing batchLP** are an important ingredient of our work and we believe they justify this research even without an end-to-end integration. Such complexity is summarized as follows: The transition from sequential to batched PDHG is conceptually simple: replace SpMV with SpMM so that $A x$ and $A^\top y$ become $A X$ and $A^\top Y$, with $X$ and $Y$ dense. Main iterations store $X$ and $Y$ in row-major format, while projections and updates require custom CUDA kernels beyond cuBLAS.
>
> The main challenges arise in the details. Periodic feasibility and optimality checks must be performed per column, requiring row/column major conversions, also used for infeasibility detection. Early termination requires extracting solutions and removing solved columns efficiently via row/column ordering transformations and custom kernels. Restart strategies must operate at the batch level, using averaged criteria that may restart some subproblems too early or late. Warm restarts are complex since subproblems differ. Efficient removal of converged columns requires batched swapping to avoid costly operations. Finally, selecting the optimal batch size involves a tradeoff between memory usage and computational efficiency. A more detailed description of those sophisticated algorithmic modifications is provided in the answer to reviewer uBJP.
>
> **W3: GPU-enabled heuristic search related work missing**
>
> Finally, in terms of GPU-based heuristics, we agree with the reviewer that we failed to discuss those in the literature review, which we will do in the final version. Prior work such as GPU A* parallelizes *tree traversal* by expanding many nodes simultaneously. Our work parallelizes the *LP solve itself* via batched matrix-matrix operations. These are complementary approaches.
>
> **Q2: Extensions to other LP tasks (cut generation, Lagrangian relaxations)**
>
> Concerning the possibility of **batching other repeated LP tasks in MIP**, we explicitly discuss in the submission the case of Optimization-based bound tightening (OBBT) and we give computational results. In the meantime, we have already figured out the potential use in solving the so-called cut-generating LP for each single-variable disjunction, so as to generate disjunctive cuts. We believe more opportunities will be available.

---

> > ### Author Rebuttal · Reviewer_yaQM · 2026-04-04
> >
> > Thank you for the clarifications.

---

### Decision · Program_Chairs · 2026-04-30

**Decision:**

Accept (regular)

**Comment:**

This paper proposes to do batched computation for a first order optimization algorithm utilizing efficient matrix matrix multiplication and thereby speeding up the solution process. While simple, multiple technical details (restarting, convergence checks etc.) complicate the overall algorithm. Very solid speedups were measured. Reviewers agree on the practical merits and the rebuttals with additional experiments helped clarify many points. We urge the authors to include the additional experiments and clarifications on the technical aspects more prominently.